# Land Use/Land Cover Changes and Their Driving Factors in the Northeastern Tibetan Plateau Based on Geographical Detectors and Google Earth Engine: A Case Study in Gannan Prefecture

**Chenli Liu** [1,2], **Wenlong Li** [1,2,*], **Gaofeng Zhu** [3], **Huakun Zhou** [4], **Hepiao Yan** [1,2] **and Pengfei Xue** [1,2]

1    State Key Laboratory of Grassland Agro-Ecosystems, College of Pastoral Agriculture Science and Technology, Lanzhou University, Lanzhou 730020, China; liuchl18@lzu.edu.cn (C.L.); yanhp19@lzu.edu.cn (H.Y.); xuepf19@lzu.edu.cn (P.X.)
2    Key Laboratory of Grassland Livestock Industry Innovation, Ministry of Agriculture, Lanzhou University, Lanzhou 730020, China
3    Key Laboratory of Western China's Environmental Systems (Ministry of Education), College of Earth and Environmental Sciences, Lanzhou University, Lanzhou 730020, China; zhugf@lzu.edu.cn
4    Key Laboratory of Cold Regions Restoration Ecology, Qinghai Province, Northwest Institute of Plateau Biology, Chinese Academy of Sciences, Xining 810008, China; hkzhou@nwipb.cas.cn
*    Correspondence: wllee@lzu.edu.cn; Tel.: +86-136-5945-1811

**Abstract:** As an important production base for livestock and a unique ecological zone in China, the northeast Tibetan Plateau has experienced dramatic land use/land cover (LULC) changes with increasing human activities and continuous climate change. However, extensive cloud cover limits the ability of optical remote sensing satellites to monitor accurately LULC changes in this area. To overcome this problem in LULC mapping in the Ganan Prefecture, 2000–2018, we used the dense time stacking of multi-temporal Landsat images and random forest algorithm based on the Google Earth Engine (GEE) platform. The dynamic trends of LULC changes were analyzed, and geographical detectors quantitatively evaluated the key driving factors of these changes. The results showed that (1) the overall classification accuracy varied between 89.14% and 91.41%, and the kappa values were greater than 86.55%, indicating that the classification results were reliably accurate. (2) The major LULC types in the study area were grassland and forest, and their area accounted for 50% and 25%, respectively. During the study period, the grassland area decreased, while the area of forest land and construction land increased to varying degrees. The land-use intensity presents multi-level intensity, and it was higher in the northeast than that in the southwest. (3) Elevation and population density were the major driving factors of LULC changes, and economic development has also significantly affected LULC. These findings revealed the main factors driving LULC changes in Gannan Prefecture and provided a reference for assisting in the development of sustainable land management and ecological protection policy decisions.

**Keywords:** Google Earth Engine (GEE); random forest; land use degree index; geographical detector; Gannan Prefecture

---

## 1. Introduction

Land use/land cover (LULC) changes are the most basic and prominent landscape characteristic describing the impact of anthropogenic disturbance on the surface of the Earth and play an important role in the studies of regional and global environmental changes [1]. In the past few decades, LULC

has undergone tremendous changes around the world, especially in the Tibetan Plateau (TP), home to Earth's highest altitude and the harshest and most sensitive environment [2,3]. The TP has warmed much faster than the global average in the past fifty years, according to observations and climate models [4,5]. In addition, increasing human population and activity seriously threaten biodiversity, ecosystem services, and wildlife habitat. Thus, monitoring LULC changes and their underlying mechanisms on the TP is vital to achieving sustainable development.

There are currently great uncertainties and differences among existing global LULC products due to different data sources, methods, and classification systems. Furthermore, most previous studies were concerned about long-term vegetation change on the TP [6,7]. The LULC of the TP is usually based on low-resolution satellite imagery, such as AVHRR and MODIS. For example, Wang et al. [3] used MODIS to quantitatively analyze the land change trends and driving factors of change on the TP from 2001 to 2015. High-resolution optical satellite images of the TP are affected by the high cloud cover and data gaps. It is a challenging task to use a single scene image to monitor LULC changes on the TP. The dense time stack method overlays all available images and replaces the area covered by clouds through stacking the coverage using another image to create a clear image [8,9]. Similar methods have been applied with good results for forest [10], land use [11], and impervious surface change monitoring [12].

However, the processing of multi-source satellite data is a huge challenge for computing capacity. Currently, a free cloud platform provides new methods for geospatial analysis. The Google Earth Engine (GEE) can avoid the process of image downloading and preprocessing and hence greatly improve the efficiency of LULC change research [13,14]. GEE provides a JavaScript and Python coding environment to facilitate data processing for the user. The application of dense time stacking of Landsat image on the GEE platform effectively overcomes the cloud cover challenge and has proven to be a successful method of solving image quality problems [9,11].

The driving force analysis can determine the process of LULC changes and reveal the regional ecological environment change mechanism [15]. The LULC changes over time and space are affected by many factors, such as society, economy, and the natural ecological environment [1,15,16]. The driving force methods of analyzing LULC changes could be divided into the qualitative description and quantitative analysis [17]. The qualitative analysis methods are coarse and can only characterize the development trend of LULC and various driving factors, and it is difficult to estimate the extent to which each factor affects LULC changes. Quantitative methods mainly use correlation analysis, multiple linear regression, principal component analysis, and logistic regression models to clarify the relationship between driving factors and LULC changes [17–19]. However, these methods are subjective and ignore the spatial relationship between driving factors and LULC changes; it is difficult to study the underlying mechanism of their inherent changes accurately. Geographical detectors are statistical methods that detect spatial differentiation and reveal influencing factors [20], doing so without too many assumptions while overcoming the limitations of traditional linear statistical methods such as correlation and regression analysis [19,21]. Thus, geographical detectors are widely used to analyze the driving factors of vegetation change [22,23] and ecosystem health [24].

Gannan Prefecture is an important water supply area for the Yellow River, located in the northeast TP. This area plays an important role in conserving water sources, maintaining biodiversity, and regulating climate [25]. To date, due to unreasonable land use and population growth, vegetation has been severely damaged, causing environmental deterioration [26]. These changes not only affect the livelihood of farmers and herdsmen but also threaten the ecological security of the Yellow River Basin and even the whole of northern China. Therefore, studying LULC changes in the Gannan Prefecture is of great significance to sustainable regional development. However, severe weather conditions make it difficult to obtain cloudless single-scene Landsat images, which limits the ability to monitor LULC changes in the area. Moreover, studies on LULC classification and its driving mechanism throughout the region are still lacking. Therefore, it is necessary to conduct a dynamic study on LULC in this area to support decision-making for economic development and ecological conservation.

This study applied dense time stacking of Landsat images to monitor LULC changes in Gannan Prefecture and explored its driving factors based on geographical detectors. We aimed to (1) obtain the LULC information of Gannan Prefecture 2000–2018 based on the GEE platform, (2) analyze the spatiotemporal change pattern of land-use intensity in Gannan Prefecture, and (3) identify the main socioeconomic and natural factors affecting LULC changes using geographical detectors. The results are expected to provide theoretical support for adjusting and optimizing land use in Gannan Prefecture.

## 2. Materials and Methods

### 2.1. Study Area

The Gannan Prefecture is within the alpine pastoral region in the southwest of Gansu Province, located in the transition zone between the northeastern Tibetan Plateau (TP) and the Loess Plateau (Figure 1). The study area covers approximately 45,000 km². The longitude ranges from 100°46′ E to 104°44′ E, and the latitude from 33°06′ N to 36°10′ N [27]. The terrain is high in the northwest and low in the southeast, with elevations varying between 1175 m and 4779 m. The study area has a typical plateau continental climate and is cold and humid. The average annual precipitation is between 400 mm and 700 mm, and the average annual temperature is below 3 °C. The region experienced significant socioeconomic development during 2000–2018, with population growth from 653,600 people to 722,951 people and GDP increasing from 137.29 to 1502.17 million yuan. Meanwhile, livestock production is the main income source in Gannan Prefecture, which promotes LULC changes to some extent [26].

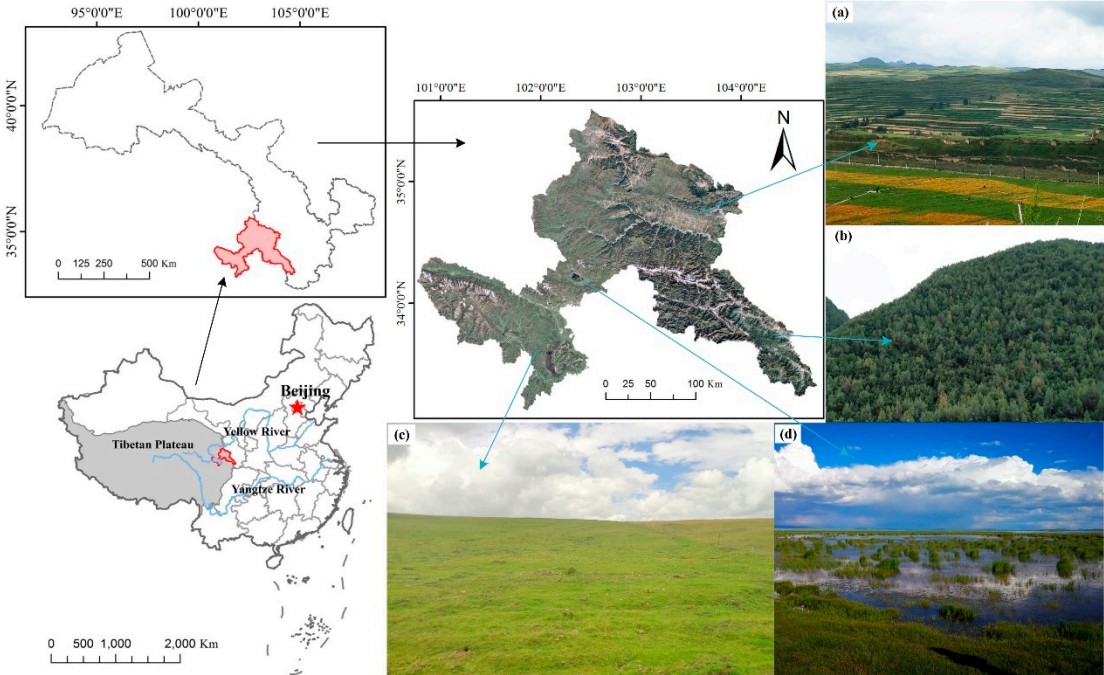

**Figure 1.** Location of Gannan Prefecture. The right column shows examples of land use/land cover (LULC) types (**a**: farmland, **b**: forest land, **c**: grassland, and **d**: wetland).

### 2.2. Data Preparation

The GEE platform provides Landsat datasets by the United States Geological Survey (USGS, https://www.usgs.gov/) [13]. All Landsat images were acquired, preprocessed, mosaicked, and processed through the JavaScript application programming interface (API). In this work, available Landsat TM and OLI Collection 1 Tier 1 top-of-atmosphere (TOA) reflectance products in Gannan Prefecture were analyzed. We choose the TOA reflectance product because its reflectance algorithm

removes the exoplanetary effects associated with variable solar irradiance as a function of variability in (1) solar zenith angles, (2) spectral band differences, and (3) Earth-to-Sun distances at different times of the year [8,28]. The elevation, slope, and aspect were extracted from a digital elevation model (DEM) with a spatial resolution of 30 m. All these images and the DEM originated from the GEE. The climate, soil, and vegetation types were obtained from the Resources and Environmental Science Data Center of the Chinese Academy of Sciences (http://www.resdc.cn/). In addition, the study also collected annual regional socioeconomic statistical data (county), including population density, gross domestic product (GDP), livestock quantity, and chemical fertilizer consumption. These data were derived from the Gansu Development Yearbook, Gansu Statistical Yearbook, and Gannan Statistical Yearbook (Table 1).

**Table 1.** Factors influencing LULC changes.

| Factors Types | Code | Index | Unit |
|---|---|---|---|
| Socioeconomic factors | $X_1$ | Population density | people/km$^2$ |
| | $X_2$ | Gross domestic product (GDP) | yuan |
| | $X_3$ | Livestock quantity | sheep |
| | $X_4$ | Chemical fertilizer consumption | ton |
| Natural factors | $X_5$ | Annual mean temperature | °C |
| | $X_6$ | Annual mean precipitation | mm |
| | $X_7$ | Elevation | m |
| | $X_8$ | Slope degree | ° |
| | $X_9$ | Aspect | ° |
| | $X_{10}$ | Vegetation type | / |
| | $X_{11}$ | Soil type | / |

### 2.2.1. Satellite Imagery

All processing of Landsat TM and OLI data was conducted on the GEE platform (https://earthengine.google.com). The image processing mainly included the following steps. (1) To correct for the problem of cloudy optical images in alpine regions, we selected all TOA reflectance data of the vegetation growth season (May–September) [6,29] for each study year. The images before and after the years were used to replace and supplement the images covered by clouds and fog, and the most available pixel image composites were produced (Supplementary Materials Table S1). (2) The clouds were removed from the Landsat images in the study area (cloud cover is less than 30%). (3) The median *ee.Reducer* function was used to generate a single image from the image collection. (4) The normalized difference vegetation index (NDVI) [30], the normalized difference built index (NDBI) [31], and the modified normalized difference water index (MNDWI) [32] were calculated for each image. (5) The slope and aspect generated from DEM data were used to enhance the classification performance. Therefore, the best image combination, cloud-free, combining NDVI, NDBI, and MNDWI, could be obtained. In order to compare with the classification results in the study, we obtained the existing global LULC products. The FROM-GLC10 product with 10 m spatial resolution was downloaded from http://data.ess.tsinghua.edu.cn [33].

### 2.2.2. Training and Validation Sample Selection

The classification system was determined based on current LULC in the Gannan Prefecture and by referring to previous research [34,35]. There were seven major LULC types: farmland, grassland, forest land, water body, wetland, construction land, and unused land (Supplementary Materials Table S2).

Supervised classification methods require high-quality training and validation samples. Training samples used to create the classification model were collected through visual interpretation and field observations [36]. Specifically, to collect the field survey points, we randomly chose points that were more than 1500 m apart to minimize spatial homogeneity [3]. The location information was acquired from the Trimble Juno series handheld GPS (https://www.trimble.com). The other reference samples were based on the high-resolution image of the Google Earth map in different years. In total,

we collected more than 1323 reference points in 2000, 2009, and 2018, covering seven LULC types (Supplementary Materials Figure S1). At least 40 samples were collected for each LULC type, 70% of which were randomly selected as participants in LULC classification as training samples, and 30% used to verify the classification results as validation samples [37].

### 2.2.3. Anthropogenic and Natural Data

Because anthropogenic and natural factors profoundly influence LULC changes, 11 potential factors were selected in this study to detect their influence on LULC changes in Gannan Prefecture in 2000, 2009, and 2018 (Figure 2). In ArcGIS 10.6 software, the natural breakpoints method was used to divide the above factors into different grades [23]. We selected the natural breakpoints method because it determines clusters according to the intrinsic attributes of the data to reduce the variance within the group and increase the variance between the groups [38]. This method has been widely used in data classification when applying the geographical detector method. Population density, GDP, livestock quantity, chemical fertilizer consumption, annual mean temperature, annual mean precipitation, and elevation were classified into six grades. The slope, aspect, and vegetation type were classified into nine classes according to previously published studies [38]. Soils were divided into 14 types. The spatial distribution maps of 11 anthropogenic and natural factors are shown in Figure 2.

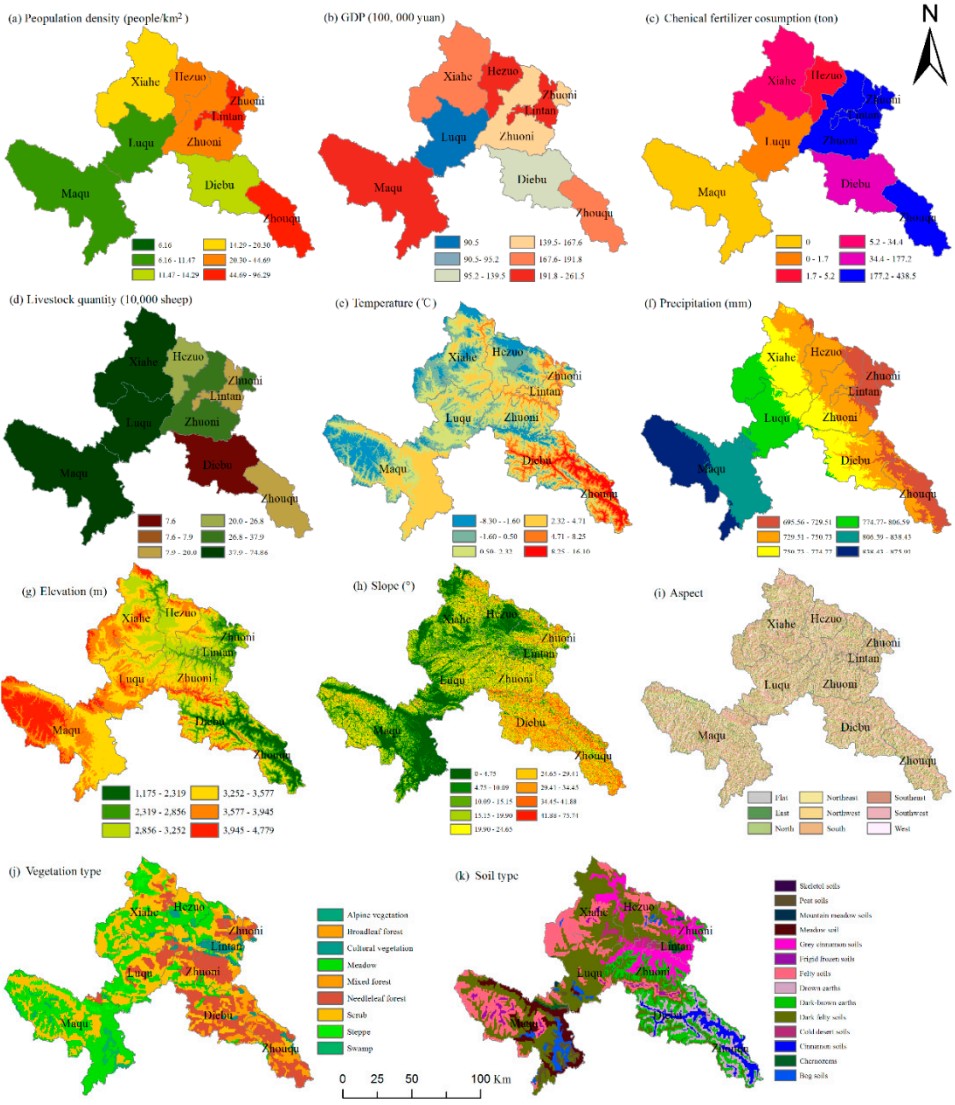

**Figure 2.** Spatial distribution maps of anthropogenic and natural components in Gannan Prefecture.

*2.3. Methods*

We used Landsat images based on the dense time stack of multi-temporal Landsat images to generate a cloudless and minimal snow cover image on the GEE platform. The specific research methods and structural framework are shown in Figure 3. First, we collected the training and validation sample datasets and uploaded them to the GEE. Landsat images were then preprocessed by date filtering, cloud masking, mosaicking, and clipping to obtain a Landsat TOA composite image and to calculate the characteristic parameters to implement the later classification. The random forest (RF) machine learning algorithm was implemented to the LULC classification, and the results were validated using a confusion matrix. Finally, the land use transfer matrix was used to analyze the change in each LULC type, and the geographical detector was used to discuss the influence mechanism of anthropogenic and natural factors on LULC changes in Gannan Prefecture.

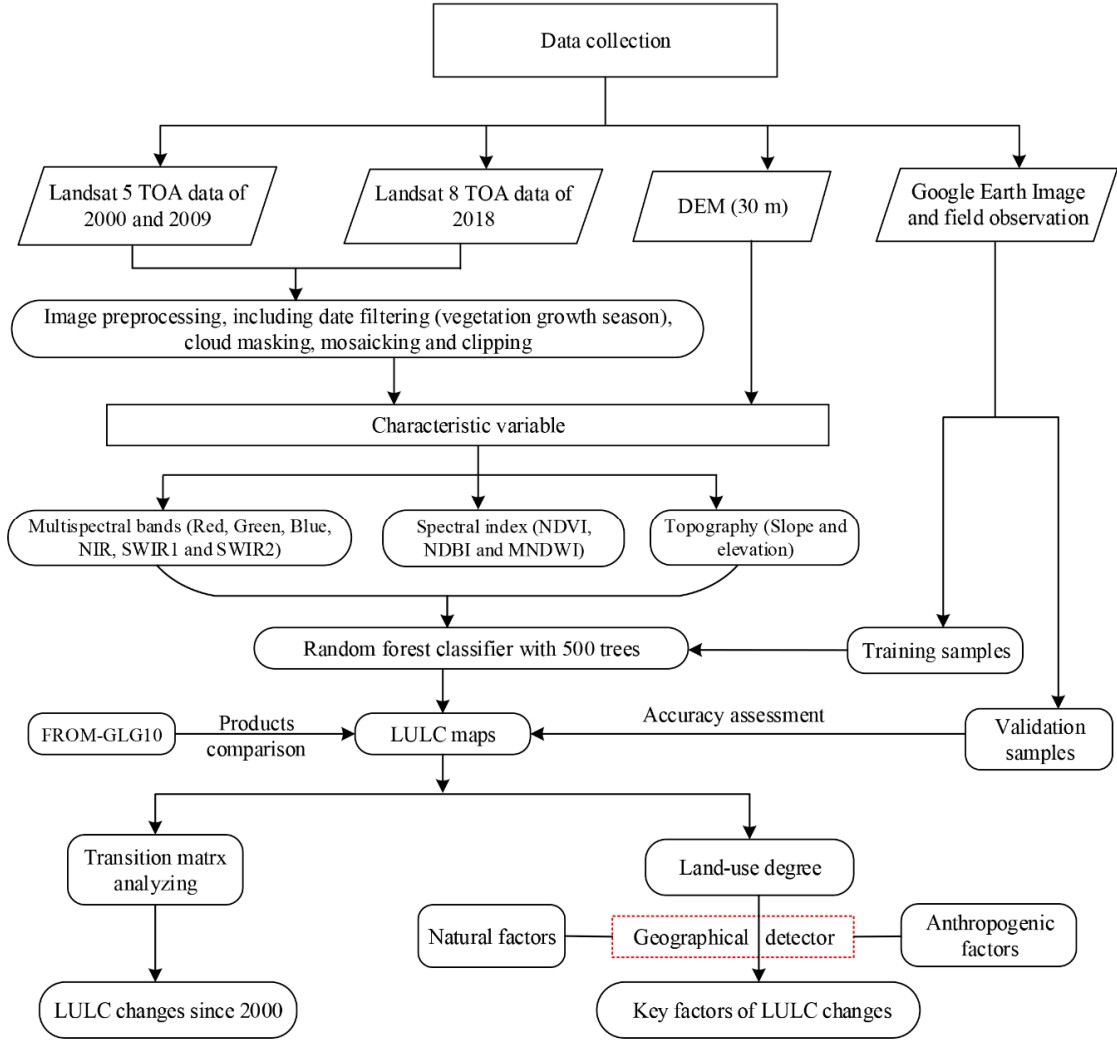

**Figure 3.** Methodology and structure framework used for LULC analysis of Gannan Prefecture.

2.3.1. LULC Classification and Accuracy Assessment

We selected a random forest (RF) algorithm as the classifier because it generally has greater processing power for data noise and overfitting [39,40]. Furthermore, RF can deal with complex data of large dimensions, and can usually provide higher accuracy than other traditional algorithms, such as maximum likelihood and single decision tree [41–43]. The RF is an ensemble learning method that creates random features and uses them to generate multiple decision trees and classifies a dataset by

using the prediction modes of all decision trees [44]. The RF classifier could easily measure the relative importance of each target variable class and has been widely applied to LULC classification due to its excellent classification results [45]. Based on the acquired samples, the *ee.Classifier.smileRandomForest* function was applied in the GEE platform to obtain the LULC classification maps for each chosen year. The RF classifier is only required to identify two parameters: the number of classification trees desired and the number of prediction variables used in each node to make the tree grow [15]. In this study, the number of trees was set to 500, and six random variables were selected from the best split when each tree grows. The application of the RF classifier in GEE could refer to the developer's guide (https://developers.google.com/earth-engine/classification).

An accuracy assessment is necessary for LULC classification to explain the correspondence between ground truth and classification results [17]. The confusion matrix is a general method for evaluating the accuracy of remote sensing image classification, which provides the correspondence between the LULC classification results and verification data. In this work, the verification of classification accuracy is reflected by overall accuracy, kappa coefficient, producer's accuracy, and user's accuracy [46].

### 2.3.2. Land Use Degree Index

Land-use intensity is the most intuitive representation of human activity and can directly reflect the state of LULC [47]. The land use degree index (*La*) directly describes the extent and intensity of land use in a specific period. Its essence is explained by the regional land use and development level, which comprehensively reflects the impact of human activities and natural environmental factors on LULC changes. The higher the value of *La*, the stronger the land-use intensity and the more complex the social and economic activities in the area [48]. *La* in Gannan Prefecture was calculated as:

$$La = 100 \times \sum_{i=1}^{n} A_i \times C_i \ La \in [100, \ 400] \tag{1}$$

where *La* is the land use degree index value; $A_i$ is the hierarchical index of land use degree *i*; $C_i$ is the percentage of the graded area of land use degree of category *i*. According to the previous studies of Zhuang et al. [48] and Liu et al. [49], the LULC types are divided into four levels and different graded indexes are assigned respectively (Table 2).

**Table 2.** The hierarchical values of LULC types in this study.

| Type of Land | Uncultivated Land | Ecological Land | Agricultural Land | Construction Land |
| --- | --- | --- | --- | --- |
| **LULC types** | Unused land (sand and bare land) | Forest land, grassland, wetland, and water body | Farmland | Urban, residential area, traffic land, and industrial land |
| **Index of Classification** | 1 | 2 | 3 | 4 |

### 2.3.3. Geographical Detector

The geographical detector model is a spatial heterogeneity detection method usually used to quantify the driving force of each factor on the dependent variable [23,24]. We used the Excel GeoDetector software developed by Wang et al. [20] to implement the geographical detector, which can be downloaded for free from the website (http://www.geodetector.cn). The geographical detector includes factor detectors, risk detectors, ecological detectors, and interactive detectors. Specifically, the factor detectors can detect whether the independent variable x is the influencing factor of variable y (land use degree index) and explain the spatial differentiation mechanism of variable y to a certain extent. Therefore, this study chose the factor detector to analyze the driving mechanisms of LULC changes, which could be measured by the value of *q*; the formula for the *q* value was estimated as:



$$q = 1 - \frac{\sum_{h=1}^{L} N_h \sigma_h^2}{N\sigma^2} \quad q \in [0,1] \tag{2}$$

where $q$ is the explanatory power index of the influencing factors of land use degree. The greater the $q$ value, the greater the effect of the independent variable x on the heterogeneity of land use. $N_h$ is the number of samples of the sub-regions $h$; $N$ is the total sample size; $\sigma$ and $\sigma^2$ denote the total variance and variance of samples in sub-region $h$, respectively.

## 3. Results

### 3.1. Variable Importance Analysis and Accuracy Assessment of LULC Classification

The RF model can analyze the importance of characteristic variables, which improves classification accuracy while reducing data redundancy and processing workload. We used the explain method on the classifier to view the importance of characteristic parameters on the GEE platform. The higher the importance score indicated, the greater the impact and contribution of the variable to the classification results. Our results showed that elevation had the highest importance score among all the characteristic variables; its average value was greater than 1100 (Figure 4). Slope, MNDWI, and NDVI followed, and they are more important for the identification of water body and vegetation. The importance score of other characteristic variables for LULC classification remained stable (Figure 4).

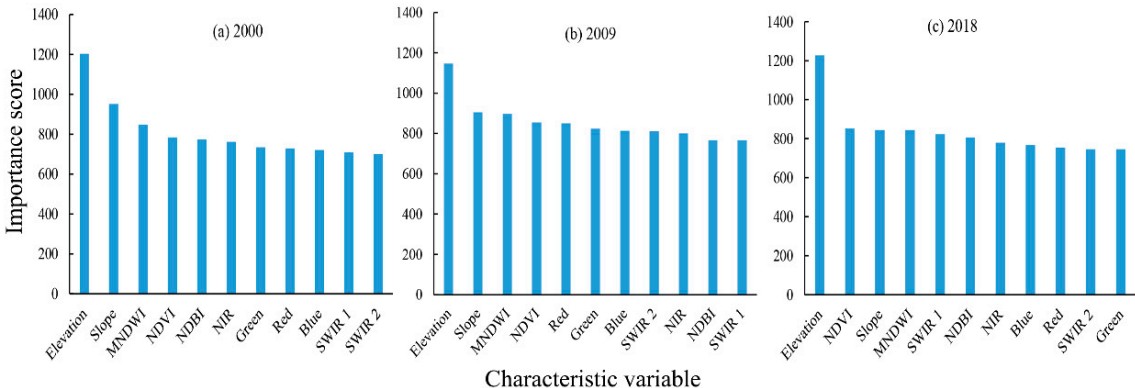

**Figure 4.** Importance distribution of characteristic variables in LULC classification.

The accuracy of the classification result is an important prerequisite for analyzing LULC changes. Our classification results showed that the overall accuracy was 90.08%, 89.54%, and 91.41% for 2000, 2009, and 2018, respectively, and the kappa coefficient was 87.63%, 87.06%, and 89.40% for 2000, 2009, and 2018, respectively (Table 3). The overall accuracy of the classification reached the acceptable threshold, indicating that the classification accuracy could meet the requirements of the LULC classification. The confusion matrix showed detailed classification accuracy for each LULC type (Table 3). Furthermore, in order to further evaluate our classification result, we compared it with FROM-GLC10 products (Figure 5). As a whole, the spatial distribution of the main LULC types was consistent through visual manual inspection. This study has a finer effect on wetland classification, but there are some misclassifications of farmland (Figure 5).

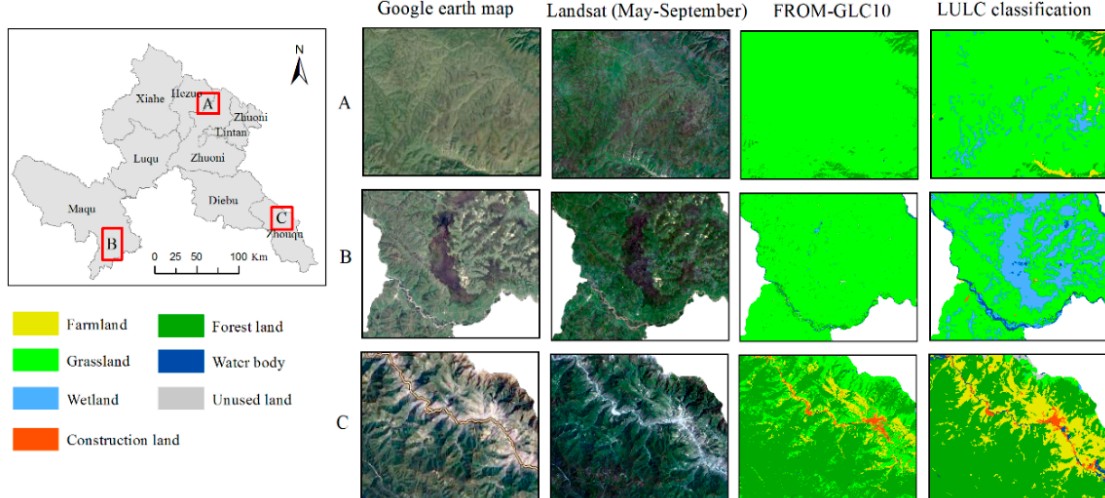

**Figure 5.** Google earth map, Landsat images in vegetation growth season, existing FROM-GLG10 products, and LULC classification results of this study.

**Table 3.** Confusion matrix of the classification. The user's accuracy, producer's accuracy, overall accuracy, and kappa coefficient for LULC classification are shown (Unit: %).

| LULC Types | 2000 | | 2009 | | 2018 | |
|---|---|---|---|---|---|---|
| | User's Accuracy | Producer's Accuracy | User's Accuracy | Producer's Accuracy | User's Accuracy | Producer's Accuracy |
| Farmland | 75 | 79.41 | 78.68 | 90.56 | 83.33 | 84.67 |
| Grassland | 87.64 | 93.97 | 90.27 | 91.54 | 90 | 84.70 |
| Forest | 99 | 98.01 | 94.62 | 94.62 | 96.77 | 97.27 |
| Water | 92.85 | 84.41 | 95.23 | 88.49 | 93.25 | 93.78 |
| Wetland | 86.48 | 82.05 | 86.66 | 92.85 | 91.89 | 82.92 |
| Construction land | 86.36 | 90.47 | 91.22 | 77.61 | 91.87 | 93.03 |
| Unused land | 88.88 | 100 | 83.33 | 83.33 | 69.23 | 90 |
| Overall accuracy | 90.35 | | 89.14 | | 91.41 | |
| Kappa coefficient | 87.97 | | 86.55 | | 89.40 | |

## 3.2. Spatiotemporal Characteristics of LULC Changes

The LULC maps of Gannan Prefecture in 2000, 2009, and 2018 are shown in Figure 6. Grassland was the main LULC type, occupying more than 55% of the total land area, and was mainly distributed in northwestern Gannan Prefecture (Maqu, Luqu, and Xiahe). The forest land was mainly concentrated in the southeastern regions of Gannan Prefecture, with an area ratio of approximately 25%. The farmland was mainly distributed in Lintan county, with approximately 10% of the total land area. The construction land area was the smallest, less than 0.71%. From 2000 to 2018, the area of forest land, construction land, and unused land expanded, especially the percentage of forest land area, which increased from 24.68% to 28.18%. However, during that period, the grassland and wetland gradually decreased (Figure 6d).

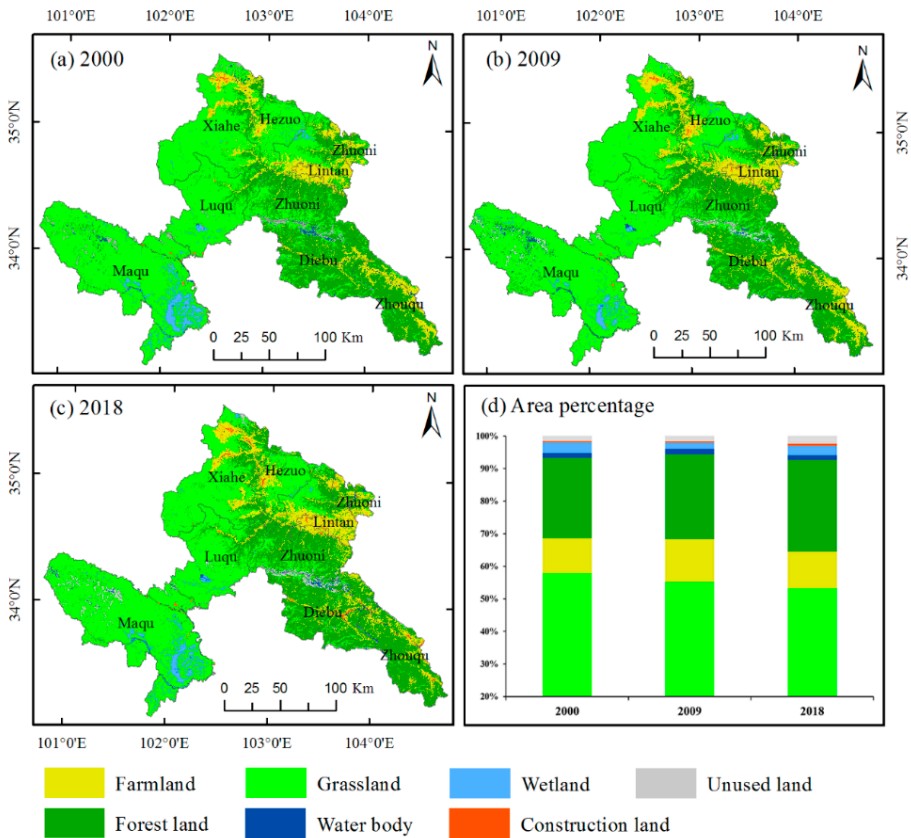

**Figure 6.** The spatial distribution and area percentage of LULC types in Gannan Prefecture in 2000–2018.

### 3.3. LULC Transformation

To observe the changes in various LULC types, we used the spatial analysis method to create the 2000–2018 LULC changes transition matrix and mapped them. From 2000 to 2009, in general, grassland and wetlands decreased, and farmland, forest land, and construction land expanded (Figures 7a and 8a). Among these, 623.44 km² and 1026.87 km² grassland was transformed into farmland and forest land, respectively. The expansion of construction land came from farmland, while the forest land was mainly converted into farmland (Figure 7a). In addition, approximately 712.15 km² of wetland was converted to grassland, and the most obvious expansion of grassland occurred in Maqu (Figure 8a).

Similarly, from 2009 to 2018, the areas of farmland and grassland continued to decrease, while forest land, wetland, and unused land expanded (Figures 7b and 8b). Specifically, farmland was mainly converted into forest land (662.06 km²), construction land (148.51 km²), and grassland (87.06 km²) (Figure 8b). Compared with the period from 2000 to 2009, the area of expanded construction land increased. The grassland continued to degrade during this period, and the grassland was mainly converted into forest land (858.14 km²), wetland (513.65 km²), and unused land (299.82 km²), which primarily occurred in Maqu, Diebu, and Zhouqu. At the same time, some of the water body area was converted to unused land.

Comparing the spatial-temporal changes in LULC between 2000–2009 and 2009–2018, we found that the farmland area increased first and then decreased, but the wetland area followed the opposite trend. In general, between 2000–2018, the grassland was continuously degraded, and the area under forest land and construction land gradually increased. The water body area did not change much, and the area of unused land also increased, which was mainly caused by grassland and water body shrinkage.

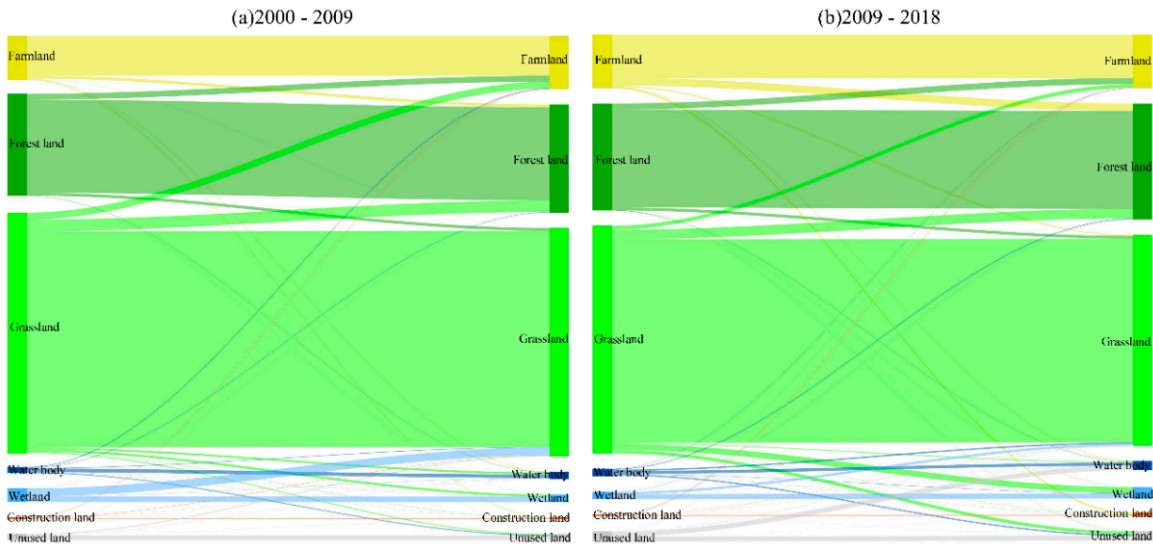

**Figure 7.** LULC changes transition matrix during (**a**) 2000–2009 and (**b**) 2009–2018. The width of the lines between the bars represents the conversion area.

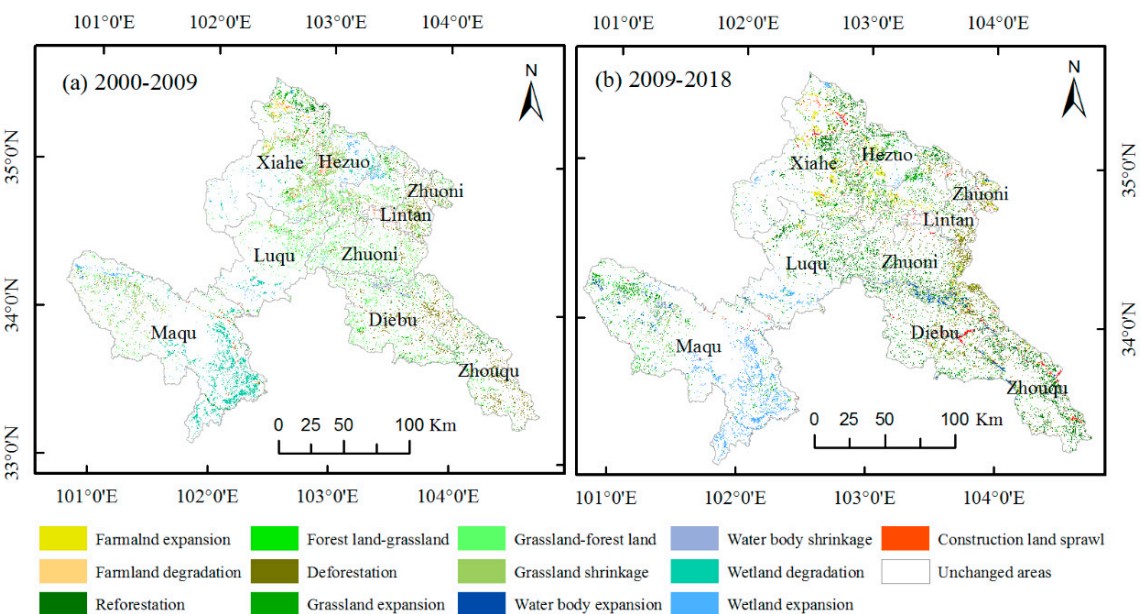

**Figure 8.** LULC transfer map of Gannan Prefecture in (**a**) 2000–2009 and (**b**) 2009–2018.

## 3.4. Degree of Land Use Change

We calculated the dynamic degree of land use based on the LULC dataset of Gannan Prefecture. The spatial changes in land-use intensity represent diversified types and multi-level intensity, and the overall land-use intensity in the northeast was higher than that in the southwest (Figure 9). The high land-use intensity was primarily concentrated in Lintan, Zhuoni, Hezuo, and the north of Xiahe, which were mainly affected by farming and other human activities. Correspondingly, the land-use intensity of Maqu, Luqu, and Zhuoni was relatively low. From these results, we detected that the most noticeable land-use intensity in Gannan Prefecture was with regard to farmland. In addition, the land-use intensity of Gannan Prefecture gradually increased between 2000–2018, but the change was slight.

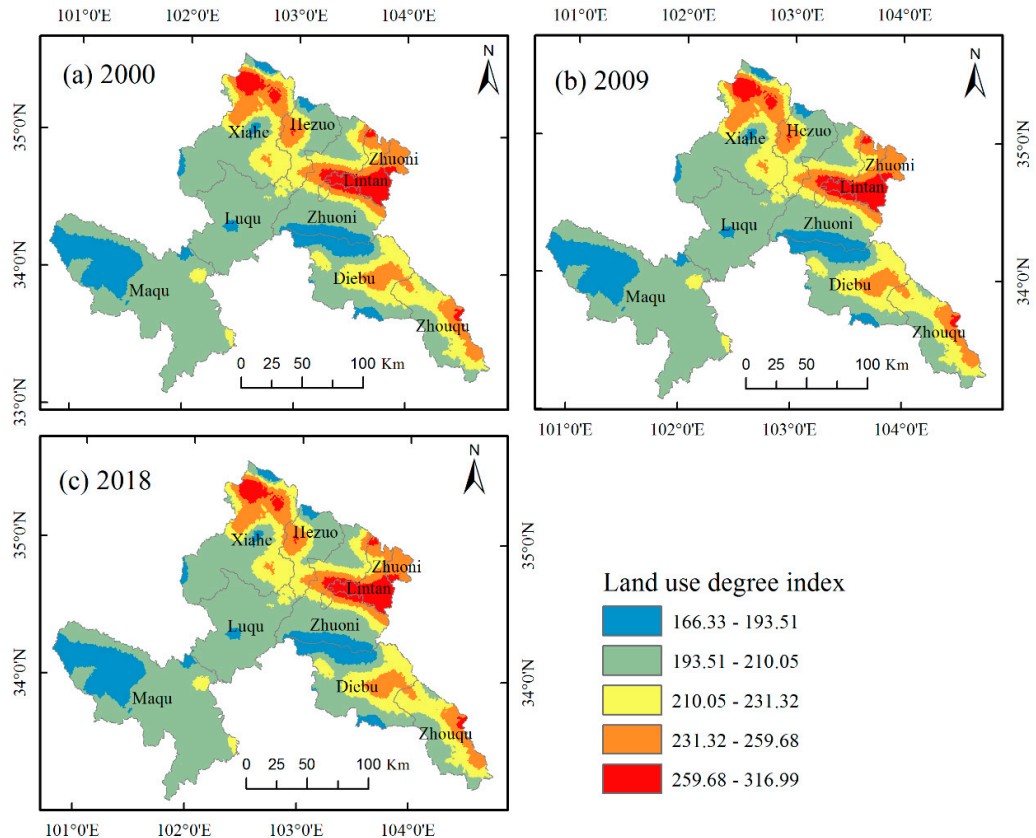

**Figure 9.** The spatial distribution of land-use intensity in Gannan Prefecture from 2000–2018. (**a**) 2000, (**b**) 2009, (**c**) 2018

### 3.5. Analysis of Driving Mechanisms in LULC Changes

Factor detection revealed the influence of each factor on land-use intensity. The calculated $q$ values of anthropogenic and natural factors (Table 4) indicate that the effect of each factor on land-use intensity was significantly different. In general, from 2000 to 2018, the $q$ values of elevation, population density, and soil types all explained the land-use intensity by more than 0.36. Therefore, elevation was the main natural factor affecting LULC changes in Gannan Prefecture. With economic developments, the $q$ values of GDP, livestock quantity, and chemical fertilizer consumption were greater than 0.1137 and demonstrated an increasing trend. In 2018, the $q$ values of these three variables were higher than 0.3117, especially for livestock quantity (0.3757), indicating that grazing is an important factor driving LULC changes. However, the influence of slope and aspect on LULC changes is very small. The $q$ value of vegetation type on LULC changes was maintained at approximately 0.36 to 0.39. In addition, each driving factor of land-use intensity was tested by significance ($p < 0.001$), except slope and aspect.

**Table 4.** Influence of anthropogenic and natural factors on LULC changes between 2000–2018.

| Year | Factor | Anthropogenic Factors | | | | | | Natural Factors | | | | |
|------|--------|-------|-------|-------|-------|-------|-------|-------|-------|-------|----------|----------|
| | | $X_1$ | $X_2$ | $X_3$ | $X_4$ | $X_5$ | $X_6$ | $X_7$ | $X_8$ | $X_9$ | $X_{10}$ | $X_{11}$ |
| 2000 | $q$ | 0.4062 | 0.2231 | 0.1930 | 0.3071 | 0.3575 | 0.0509 | 0.4689 | 0.0062 | 0.0041 | 0.2690 | 0.3848 |
| | $p$ value | 0.000 | 0.000 | 0.000 | 0.000 | 0.000 | 0.000 | 0.000 | 1 | 1 | 0.000 | 0.000 |
| 2009 | $q$ | 0.3796 | 0.2601 | 0.1137 | 0.3904 | 0.3103 | 0.2432 | 0.4486 | 0.0083 | 0.0029 | 0.2572 | 0.3604 |
| | $p$ value | 0.000 | 0.000 | 0.000 | 0.000 | 0.000 | 0.000 | 0.000 | 1 | 1 | 0.000 | 0.000 |
| 2018 | $q$ | 0.3740 | 0.3117 | 0.3757 | 0.3731 | 0.2077 | 0.4013 | 0.4544 | 0.0086 | 0.0023 | 0.2780 | 0.3952 |
| | $p$ value | 0.000 | 0.000 | 0.000 | 0.000 | 0.000 | 0.000 | 0.000 | 1 | 1 | 0.000 | 0.000 |

Note: the $q$ values indicate the explanatory power of each factor on LULC changes.

## 4. Discussion

### 4.1. Analysis of LULC Classification Variable Importance and Result Verification

This study constructed various characteristic variables on the GEE platform and used the RF machine algorithm to classify LULC in Gannan Prefecture. The RF algorithm provides flexibility to include different data types in the modeling process and accurately classifies the LULC heterogeneity [50,51]. The variable importance analysis shows that elevation makes the greatest contribution to LULC classification (Figure 4), perhaps because of the high elevation and complex topography of the area, which has a significant impact on the spatial distribution of precipitation, temperature, and vegetation types [25]. The slope can be significant for the spatial distribution of farmland, water body, and wetlands. Hoshikawa et al. [52] also found that terrain has a significant influence on LULC classification. The MNDWI and NDVI are accurate and widely used in extracting water bodies and vegetation [53,54].

The overall accuracy and kappa coefficient were reasonable, as, according to the USGS survey, the recommended threshold is 85%, and Gashaw et al. [55] recommends 80%. However, we found errors in LULC classification in some areas through manual visual interpretation. Accurate training samples and validation samples are very important for classification accuracy [56], so the classification errors in our results are partly due to impure training samples capturing mixed LULC types. From 2000 to 2018, according to the users' accuracy and producers' accuracy of LULC classification results, the classification accuracies of grassland and forest land were the highest. This may be because the grassland and forest land in the study area are relatively concentrated. To further validate the study, our results were compared with those of Gong et al. [35], which were found to be consistent with our results; in particular, the wetland classification effect of this study is more refined. However, some farmland and grassland areas were misclassified, which may be caused by grassland degradation, resulting in similar spectral characteristics between farmland and grassland, and generates "salt-and-pepper" effects, which mainly occurred in Xiahe county.

### 4.2. Temporal-Spatial Variation of LULC Changes

LULC changes are influenced by the interaction between humans and the environment at different spatial and temporal scales. Monitoring LULC changes can help us understand the causes of their dynamic changes, and it supports land management and decision-making [57]. Our LULC classification results indicated that grassland was the main LULC type in Gannan Prefecture, followed by forest land and farmland, which was consistent with other existing global LULC products [35]. Comparing the LULC changes in Gannan Prefecture in the periods of 2000–2009 and 2009–2018, the change trajectories of the main LULC types in these two periods were the same. The grassland has continued to decline, but the forest land shows the opposite trend. The farmland area expanded broadly before 2009 and decreased after 2009. The difference can probably be attributed to the continuous implementation of the Project of Grain for Green. The expansion of construction land was faster after 2009, which might be due to the national development strategy promoting the accelerated development of western China [34]. Moreover, the wetland area increased from 2009 to 2018, which was mainly due to the increase in annual mean precipitation, but there was still a decreasing trend compared with 2000.

The change in land-use intensity reflects the disturbance degree of human activities on natural resources [58]. The land-use intensity of Gannan Prefecture has significant spatial differentiation characteristics. The high land-use intensity is mainly concentrated in the northwest of Gannan Prefecture, in which there are transition zones for agriculture, pastoral and forest areas, the terrain is relatively flat, and the climate is suitable, which is conducive to the growth of crops, especially in Lintan. These areas are dominated by farmland and construction land, so the regional land exploitation is relatively high. However, the southwestern part of Gannan Prefecture is high in elevation, and people's livelihoods depend largely on grazing Tibetan sheep and yaks, so the degree of land use development is low.

### 4.3. Major Drivers of LULC Changes

LULC is a complex process affected by natural, social, and economic factors. Here, we explain the driving mechanism of LULC changes by analyzing the relative importance of natural and human disturbances. Generally, LULC changes are greatly affected by human activities. Our results showed that from 2000 to 2018, population density explained much of the LULC changes. This finding agrees with previous reports that the LULC changes driven by human activities are increasing [19]. Furthermore, economic development significantly changed the LULC in Gannan Prefecture. Wang et al. [3] found that the rapid development of tourism and infrastructure has attracted an increasing number of agricultural workers to the urban areas on the TP. Similar to many other regions in China, construction land expansion is mainly due to the conversion from farmland near the major cities [36]. Luan et al. [59] also found that population growth significantly increased the rapid expansion of Pan-Third Pole cities around the TP. Unlike other developed regions, livestock grazing activities are the most common factor influencing LULC changes in Gannan Prefecture [18]. The factor detection showed that the impact of livestock quantity on LULC changes in 2018 was second only to elevation. This result indicates that grassland degradation is relatively severe where overgrazing and social productivity are high [60]. In addition, Meng et al. [25] reported that the grassland coverage in Xiahe has declined in the past 17 years.

LULC changes are also closely related to natural factors [18]. Our results showed that from 2000 to 2018, elevation was the most important natural factor, possibly because in alpine regions, elevation determines human activities and affects the LULC pattern. The high elevation limits the urbanization process in this area and generates an unsustainable land expansion model [59]. In addition, Wang et al. [61] reported that topography affects the LULC classification of the Qilian Mountains. The change in environmental conditions can control the process of LULC changes [62]. In our study, the influence of vegetation and soil type on LULC change remained relatively stable, with explanatory powers of 0.26 and 0.36, respectively. These results highlighted that the soil type in alpine regions is difficult to change significantly in the short term, and that climate change is the main factor affecting the distribution of vegetation in Gannan Prefecture [25]. Specifically, our results showed that the effects of precipitation and temperature on LULC changes were different. From 2000 to 2018, the impact of precipitation on LULC changes gradually increased, whereas the temperature showed the opposite trend, especially in the wetland area. Wang et al. [63] similarly proposed that the significant increase of wetland areas was primarily driven by the positive precipitation trend on the TP.

### 4.4. Advantages and Limitations of the Current Study

The GEE cloud computing platform has unique computing capabilities, free satellite images, integrated APIs, and researchers can process massive geographic data [14]. In addition, the dense time stack of Landsat images can solve the problems in analyzing the data from cloudy alpine regions. Based on GEE, the image cloud removal method and the training sample can quickly and accurately be applied to the Gannan Prefecture LULC classification. Although GEE contains a large number of image archives, there are some computational limitations when processing images. For example, in some cases, users will encounter internal problems when performing a large number of data calculations, such as calculation timeout, exceeding user memory, and limitations of the output size [44].

Factor detection found that the influence of natural factors on LULC changes in Gannan Prefecture is greater than the influence of the anthropogenic factors, which might be caused by the fragile ecological environment and lack of investment and transportation [18]. However, with economic development, the contribution of socioeconomic factors to LULC changes has increased. In addition, the statistical data in this study are spatialized using the assignment method, which leads to the problem of data uniformity in the spatial unit. Therefore, it is essential for the spatialization of socioeconomic data in the future to establish a comprehensive model that integrates multi-source data.

## 5. Conclusions

The study used the dense time stacking of multi-temporal Landsat images and the RF machine learning algorithm to map the LULC in Gannan Prefecture, and then analyze potential driving forces based on geographic detectors. Our results demonstrated that from 2000 to 2018, Gannan Prefecture was dominated by grassland, followed by forest land and farmland. Grasslands and wetlands gradually degraded from 2000 to 2018, while forest land, construction land, and unused land expanded. Grassland was mainly converted to forest land. The high land-use intensity was distributed in Lintan, northern Xiahe, and Hezuo. The overall land-use intensity of Gannan Prefecture was higher in the northeast than in the southwest, and these areas are relatively flat, mainly consisting of cultivated land and construction land. In addition, the importance analysis of all variables using the RF model found that elevation was the most important characteristic variable for LULC classification in Gannan Prefecture.

The driving factor analysis also found that elevation was the most important factor affecting the LULC in Gannan Prefecture. However, with economic development, the driving effects of anthropogenic factors on the LULC have gradually expanded. In 2018, population density, GDP, livestock quantity, and chemical fertilizer consumption had an explanatory power exceeding 0.3117 for LULC changes, which indicates that economic development has promoted LULC changes. In conclusion, our study results indicate that elevation restricts the dominant factors of LULC changes in alpine regions. The study provides results that revealed the land use mechanism in the northeastern TP.

**Supplementary Materials:** The following are available online at http://www.mdpi.com/2072-4292/12/19/3139/s1, Figure S1: Map of the reference training and validation data of Gannan Prefecture collected using Google Earth high spatial resolution imagery, Table S1: Images comprising each study year and the previous and subsequent years composite, Table S2: Definition of land use classes of classification scheme adopted in this study, Text S1: JavaScript code used within GEE for image classification.

**Author Contributions:** C.L. and W.L. had the original idea and designed this study. H.Y. and P.X. were responsible for data processing and analysis. G.Z. and H.Z. contributed to the revision of the manuscript. Writing—review & editing, W.L. and C.L. All authors have read and agreed to the published version of the manuscript.

**Funding:** This research was funded by the National Key Research and Development Program of China (2018YFC0406602, 2016YFC0501901), the NSFC grants(41471450, 31170430), The 2019 Opening Project of the Ecological Research Center of the Northwest Plateau Institute of Biology of Chinese Academy of Sciences, the Second Tibetan Plateau Scientific Expedition and Research Program (2019QZKK0302), and the Fundamental Research Fund for the Central Universities (lzujbky-2019-it03).

**Acknowledgments:** We thank the Google Earth Engine platform and developers for their support. We also thank the journal editor and the anonymous reviewers for their useful comments and great efforts on this paper.

**Conflicts of Interest:** The authors declare no conflict of interest.

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
