# Peer review of "Land Use/Land Cover Changes and Their Driving Factors in the Northeastern Tibetan Plateau Based on Geographical Detectors and Google Earth Engine: A Case Study in Gannan Prefecture"

_remotesensing, doi:10.3390/rs12193139_

Round 1

Reviewer 1 Report

The revised manuscript met my revision requirements. I suggest accepting it after minor revision.

Author Response

Reviewer #1:

The revised manuscript met my revision requirements. I suggest accepting it after minor revision.

Response: Thank you for the constructive comment. We have checked the English language problems once more, and hope that it now meets the journal’s standard. We have sought an English editing service from native English speaker and attach a manuscript editing certificate of editing (EDITORIAL CERTIFICATE).

Reviewer 2 Report

The authors provided a suitable reply to my comments. They did a serious job and worked very hard in clarifying their view, goals of this article, and bias analysis. I deeply respect them.

However, I'm still convinced that this article is offering a case study not so aligned with the scientific impact of Remote Sensing, where novel articles, not too narrow in their study areas, are welcome. Remote Sensing is seeking really high impact articles where or a new methodology is presented or large scale analysis are discussed to understand big challenges. 

I'm sorry, but from my side and according to my experience in science and editorial duties, I cannot propose to accept this article. I'm pretty sure that the authors will find another remote sensing journal. 

Author Response

Reviewer #2:

The authors provided a suitable reply to my comments. They did a serious job and worked very hard in clarifying their view, goals of this article, and bias analysis. I deeply respect them.

However, I'm still convinced that this article is offering a case study not so aligned with the scientific impact of Remote Sensing, where novel articles, not too narrow in their study areas, are welcome. Remote Sensing is seeking really high impact articles where or a new methodology is presented or large scale analysis are discussed to understand big challenges.

I'm sorry, but from my side and according to my experience in science and editorial duties, I cannot propose to accept this article. I'm pretty sure that the authors will find another remote sensing journal.

Response: Thank you for this comment. Remote Sensing publishes regular research papers, reviews, letters and communications covering all aspects of remote sensing science. We believe that this manuscript is appropriate for publication by the Journal of Remote Sensing because it focuses on remote sensing applications. In addition, a number of scholars also published many articles on land cover and land use (LULC) mapping in the journal in recent years. For example, (1) Monitoring and Projecting Land Use/Land Cover Changes of Eleven Large Deltaic Areas in Greece from 1945 Onwards (Krina et al, 2020); (2) Quantitatively Assessing and Attributing Land Use and Land Cover Changes on China's Loess Plateau (Du et al, 2020). (3) Land Cover Changes and Their Driving Mechanisms in Central Asia from 2001 to 2017 Supported by Google Earth Engine (Hu et al., 2019); (4) Mapping Vegetation and Land Use Types in Fanjingshan National Nature Reserve Using Google Earth Engine (Tsai et al, 2018); (5) Monitoring of Land Use/Land-Cover Changes in the Arid Transboundary Middle Rio Grande Basin Using Remote Sensing (Mubako et al, 2018).

However, compared to these studies, due to the complex terrain and poor remote sensing image quality in this study area, there are great uncertainties in existing LULC products. Furthermore, most previous studies were concerned about the long-term change of vegetation in the Tibetan Plateau. To date, there is still a lack of study on LULC classification in the Tibetan Plateau, and the LULC is usually based on low-resolution satellite imagery, such as AVHRR and MODIS. For example, Wang et al. (2019) quantitatively analyzed the land change trends and driving factors on the TP from 2001 to 2015 based on MODIS. The high-resolution optical satellite images of the Tibetan Plateau are affected by the high cloud cover and data gaps. It is a challenging task to use a single scene image to monitor LULC changes on Tibetan Plateau. Furthermore, the driving force analysis of LULC changes currently mainly uses traditional linear statistical methods such as correlation and regression analysis. However, these methods are subjective and ignore the spatial relationship between driving factors and LULC changes, it is difficult to accurately study the underlying mechanism of their inherent changes. Therefore, this study applies geographical detectors to detect spatial differentiation and influencing factors of LULC changes.

Given the discussions above, this study applies dense time stacking of Landsat image on the GEE platform to effectively overcome the cloud cover challenge, which has proved to be a successful method to solve image quality problems. Then, the RF model is used to analyze the importance of characteristic variables, which could improve classification accuracy when reducing data redundancy and processing workload. Finally, geographical detectors are used to quantify the impact of driving factors on LULC changes. This method considers the unique spatial differentiation of spatial data and quantifies the contribution of various influencing factors to LULC change, which could effectively explain how each variable influences the spatial differentiation of LULC changes.

In addition, we further evaluate the accuracy of the classification results of the proposed method and compared them with the existing FROM-GLC10 products (lines 303-306 and Figure 5). As a whole, the spatial distribution of the main LULC types was consistent through visual manual inspection, especially the classification of wetland in this study is more refined. We have also discussed in detail the accuracy and deviation of the final classification results (lines 404-409). For LULC transformation change, we use Sankey Diagram to intuitively reflect the changes in LULC in Gannan Prefecture in 2000–2018 (Figure 7 and lines 332-334). We believe that the results of this study can provide an important scientific basis for LULC monitoring and its mechanism analysis in the northeastern Tibetan Plateau.

References cited in the responses:

  1. Krina, A.; Xystrakis, F.; Karantininis, K.; Koutsias, N. Monitoring and Projecting Land Use/Land Cover Changes of Eleven Large Deltaic Areas in Greece from 1945 Onwards. Remote Sens. 2020, 12, doi:10.3390/rs12081241.
  2. Du, X.Z.; Zhao, X.; Liang, S.L.; Zhao, J.C.; Xu, P.P.; Wu, D.H. Quantitatively Assessing and Attributing Land Use and Land Cover Changes on China's Loess Plateau. Remote Sens. 2020, 12, 18, doi:10.3390/rs12030353.
  3. Hu, Y.F.; Hu, Y. Land Cover Changes and Their Driving Mechanisms in Central Asia from 2001 to 2017 Supported by Google Earth Engine. Remote Sens. 2019, 11, 21, doi:10.3390/rs11050554.
  4. Tsai, Y.H.; Stow, D.; Chen, H.L.; Lewison, R.; An, L.; Shi, L. Mapping Vegetation and Land Use Types in Fanjingshan National Nature Reserve Using Google Earth Engine. Remote Sens. 2018, 10, 14, doi:10.3390/rs10060927.
  5. Mubako, S.; Belhaj, O.; Heyman, J.; Hargrove, W.; Reyes, C. Monitoring of Land Use/Land-Cover Changes in the Arid Transboundary Middle Rio Grande Basin Using Remote Sensing. Remote Sens. 2018, 10, 17, doi:10.3390/rs10122005.
  6. Wang, C.; Gao, Q.; Yu, M. Quantifying Trends of Land Change in Qinghai-Tibet Plateau during 2001-2015. Remote Sens. 2019, 11, 21, doi:10.3390/rs11202435.

Reviewer 3 Report

I think the authors have addressed my comments and suggest the publication of this paper. I'd still suggest the authors to further improve the language of this paper - in particular, the newly added paragraph (line 263-271) has several grammar problems.

Author Response

Reviewer #3:

I think the authors have addressed my comments and suggest the publication of this paper. I'd still suggest the authors to further improve the language of this paper. In particular, the newly added paragraph (line 263-271) has several grammar problems.

Response: Thank you for the constructive comment. We have further improved the manuscript in terms of language and readability, and attach a manuscript editing certificate of editing (EDITORIAL CERTIFICATE). We hope that it now meets the journal’s standard.

Round 2

Reviewer 2 Report

Dear authors,

the new reply you provided with the justification of the novel aspect respect to the recent literature, could be suitable. I suggest to provide an additional description this point in the text, in order to give to the readers a clear report on what is novel here to the recent literature on the same topic.

The paper then could have a window for publication.

Author Response

Thank you again for your letter and the comments concerning our manuscript entitled “Land use/land cover changes and their driving factors in the northeastern Tibetan Plateau based on geographical detectors and Google Earth Engine: A case study in Gannan Prefecture” (No: remotesensing-938989). These comments are all valuable and very helpful for improving our work and revising the manuscript. Based on the last revision, we have studied the comments carefully and revised the manuscript. This letter is our point-by-point response to the comments raised by the reviewers. The revisions are marked in red in the manuscript. The responses to the reviewers are as follows:

Reviewer #2:

The new reply you provided with the justification of the novel aspect respect to the recent literature, could be suitable. I suggest to provide an additional description this point in the text, in order to give to the readers a clear report on what is novel here to the recent literature on the same topic.

Response: We really appreciate the reviewer’s instructive suggestions for our manuscript. Based on this comment, we revised our introduction (lines 51-60, 79-84 and 94-97). In addition, we have checked the English language problems once more, and hope that it now meets the journal’s standard. We have sought an English editing service from native English speaker and attach a manuscript editing certificate of editing (EDITORIAL CERTIFICATE).

This manuscript is a resubmission of an earlier submission. The following is a list of the peer review reports and author responses from that submission.

Round 1

Reviewer 1 Report

The paper is well written in general and represent a significant amount of work for the classification and validation of the land cover/land use maps of 2000, 2009 and 2018 in the northeast Tibetan Plateau. The study also gives an interesting view of land cover/land use changes in the northeast Tibetan Plateau linked with driving factors data (e.g., topography, demography, and soil types). The aim of the paper is clear to provide information about land cover/land use change in an area that is a challenge to monitor with single scene imagery due to persistent cloud coverage. However, there are many issues that need to be addressed before considering publish.

The authors may misuse "land use" as "land cover". Land use indicates how people use the land (i.e., whether for development, conservation, or mixed uses) whereas land cover documents the physical land type such as forest or open water. We generally use the synthesis term "land cover/land use" in most cases except some specific studies which focused solely on one term.

The title should be improved, for instance, "northeast Tibetan Plateau" is a large area, but this study just focused on a small local area in southwestern Gansu province. The term "dense time stack of Landsat images" usually used when performing time-series analysis. But in this study, the authors just generated a cloud-free composited image based on multi-temporal Landsat images instead of implementing a time-series analysis.

The abstract could be improved to present better the results of the study, especially on the topics of driving factors analysis.

L47-59 the literature review is little bit superficial, the authors did not put forward a clear logic from many documented case studies about regional land cover/land use over similar study areas.

L81 Please review more literature about traditional statistical methods.

L82-84 it should appear in the section of “Method” or “Results” instead of “Introduction”.

L88 "unreasonable land use" includes "over-grazing"

L93 "cloudless Landsat images". I would say "cloudless single-scene Landsat image"

Figure 1. Why do Figure 1a and 1b show Google high-resolution imagery while 1c and 1d show photos?

L125 Could you explain why you choose to use TOA instead of surface reflectance, where both products are available? Are there some hypotheses for use TOA to get a better result?

L141 Could you be more specific with the definition of the growth season = any acquisition from 1st of May to 31st of September?

L157-188 Carefully preparing references samples is the key to get accurate classifications. What are the criterions when collecting the references data? Please give some detailed explanations in the main context instead of putting them in the supplementary text.

Section 2.2.1

How did you keep Landsat composited image consistent between Landsat 5 and Landsat 8?

Section 2.3.1,

How did you training RF model and apply the trained model on composited images? Details of Random forest tuning are also needed. The most important variables should also be described and discussed.

L166-167 why did you use "natural breakpoints method"?

L171 "map" should be "maps"

Table 2 what is the difference between artificial grassland and natural grassland? Did you classify them in your classification maps? If yes, how about classification accuracy of them?

Section 2.3.3 How did you implement the geographical detector method, with R package or what else?

Figure 4. I would rather see the full error matrices rather than Figure 4.

Figure 5. Compared with Figure 5a and 5b, there is a large expansion of construction land appeared in Figure 5c. Have you misclassified these patches since it is unreasonable to see such a large patch of urban expansion here?

It is difficult to understand Figure 6 without reading the text, please synthesis and simplify it to some specific topics instead of displaying all the numbers.

Figure 6 shows a lot of salt and pepper effect - which of these changes are real and what are the false changes?

L281-286 Is nearly-ten year a reasonable time span to monitor the changes in land cover and land uses? Could you comment on that, especially when you are talking about the effect of grassland degradation may be caused by drought?

In discussion section, can you discuss about which land cover/land use classes are better classified? Also linked with the number of samples available from each class to train the classification.

L208 and L335 Estimating intensity of land cover/land use changes based on two dates is not reliable, how did you comment on this, please add it in the discussion section?

The discussion part is a little bit superficial without deep discussion about the driving factors of land cover/land use change. For instance, did these changing cases occur here similar as (or different from) other areas of the northeast of Tibetan Plateau? And how are these changes in other studies emphasized the similar drivers or different drivers, such as the economic shift and the rural-to-urban migration? And how were your results supported by other references? Cite references. The articles below focused on land cover/land use changes and drivers, may be good sources of evidence to enhance your discussion part:

1) https://dx.doi.org/10.1002/ldr.3312

2) https://doi.org/10.3390/rs11202435

3) https://dx.doi.org/10.3390/su9091539

4) https://doi.org/10.1038/srep37658

5) https://dx.doi.org/10.1016/j.rse.2012.02.010

6) https://dx.doi.org/10.1007/s00267-013-0139-0

Author Response

Dear Editors and Reviewers:

       Thank you for your letter and the comments concerning our manuscript entitled “Land use changes and driving factors in the northeast Tibetan Plateau: analysis based on a dense time stack of Landsat images and a geographical detector” (No: remotesensing-876172). These comments are all valuable and very helpful for improving our work and revising the manuscript. We have studied the comments carefully and revised the manuscript cogitatively. This letter is our point-by-point the details of the response to the comments raised by the reviewers’ comments. The revisions are marked in red in the manuscript. The responses to the reviewers are as follows:

Reviewer #1:

        The paper is well written in general and represent a significant amount of work for the classification and validation of the land cover/land use maps of 2000, 2009 and 2018 in the northeast Tibetan Plateau. The study also gives an interesting view of land cover/land use changes in the northeast Tibetan Plateau linked with driving factors data (e.g., topography, demography, and soil types). The aim of the paper is clear to provide information about land cover/land use change in an area that is a challenge to monitor with single scene imagery due to persistent cloud coverage. However, there are many issues that need to be addressed before considering publish.

  1. The authors may misuse "land use" as "land cover". Land use indicates how people use the land (i.e., whether for development, conservation, or mixed uses) whereas land cover documents the physical land type such as forest or open water. We generally use the synthesis term "land cover/land use" in most cases except some specific studies which focused solely on one term.

Response: We agree with the reviewer on this point and have modified the entire manuscript according to the suggestion.

  1. The title should be improved, for instance, "northeast Tibetan Plateau" is a large area, but this study just focused on a small local area in southwestern Gansu province. The term "dense time stack of Landsat images" usually used when performing time-series analysis. But in this study, the authors just generated a cloud-free composited image based on multi-temporal Landsat images instead of implementing a time-series analysis.

Response: We really appreciate the reviewer’s instructive suggestions on the title of our manuscript. We have changed the title to “Land use/Land cover changes and its driving factors in the northeastern Tibetan Plateau based on geographical detectors and Google Earth Engine: A case study in Gannan Prefecture”.

  1. The abstract could be improved to present better the results of the study, especially on the topics of driving factors analysis.

Response: We really appreciate the reviewer’s instructive suggestions for our manuscript. Based on this comment, we revised our abstract (lines 33-36)

  1. L47-59 the literature review is little bit superficial, the authors did not put forward a clear logic from many documented case studies about regional land cover/land use over similar study areas.

Response: This suggestion is very important for our paper revision. Based on this suggestion, we rewrite this part of the content (lines 44-61).

  1. L81 Please review more literature about traditional statistical methods.

Response: Thank you for point this out, we have added descriptions and references of traditional statistical methods (line 86).

  1. L82-84 it should appear in the section of “Method” or “Results” instead of “Introduction”.

Response: We have modified it according to the comment.

  1. L88 "unreasonable land use" includes "over-grazing".

Response: This part has been changed according to the comment (lines 91-92).

  1. L93 "cloudless Landsat images". I would say "cloudless single-scene Landsat image".

Response: Changed as requested (line 96).

  1. Figure 1. Why do Figure 1a and 1b show Google high-resolution imagery while 1c and 1d show photos?

Response: Thank you for pointing this out. Based on our field survey, Figure 1a and 1b have been revised (line 124).

  1. L125 Could you explain why you choose to use TOA instead of surface reflectance, where both products are available? Are there some hypotheses for use TOA to get a better result?

Response: Thank you for pointing this out. We used the TOA reflectance product due to the reflectance algorithm removes the exoplanetary effects associated with variable solar irradiance as a function of variability in (1) solar zenith angles, (2) spectral band differences, and (3) Earth-to-Sun distance at different times of the year (lines 131-134).

  1. L141 Could you be more specific with the definition of the growth season = any acquisition from 1st of May to 31st of September?

Response: Thank you for pointing this out. The definition of the growing season is based on our multiyear on alpine grasslands on the Tibetan Plateau [1]. In addition, Ran et al. [2] and Wang et al. [3] also found that the growing season is from May to September on the Tibetan Plateau.

  1. L157-188 Carefully preparing references samples is the key to get accurate classifications. What are the criterions when collecting the references data? Please give some detailed explanations in the main context instead of putting them in the supplementary text.

Response: Following this comment, we have added criterions for collecting sample data (lines 165-173).

  1. Section 2.2.1 How did you keep Landsat composited image consistent between Landsat 5 and Landsat 8?

Response: Thank you for this comment. The Landsat 5 composite images were mainly used for land classification research in 2000 and 2009, while the Landsat 8 composite images were mainly used for land classification research in 2018. Therefore, this study did not composite Landsat 5 and Landsat 8 into one image.

  1. Section 2.3.1 How did you training RF model and apply the trained model on composited images? Details of Random forest tuning are also needed. The most important variables should also be described and discussed.

Response: Following this comment. In this study, the ee.Classifier.smileRandomForest function was applied in GEE platform to obtain the LULC classification maps for each chosen year. The prediction model of the RF classifier only requires two parameters to be identified: the number of classification trees desired, and the number of prediction variables, used in each node to make the tree grow. In this study, we found that the number of trees was set to 500 trees, six random variables to be selected from the best split when grows each tree (lines 217-222).

In addition, we added analysis and discussion of the importance of characteristic variables. We use the explain method on the classifier to view the importance of characteristic parameters on GEE platform. The higher the importance score indicated that the greater the impact and contribution of the variable to the classification results. We found that elevation has the highest importance score among all the characteristic variables, with an average of more than 1100 (lines 262-272).

  1. L166-167 why did you use "natural breakpoints method"?

Response: The natural breakpoints method built in ArcGIS determines clusters according to the intrinsic attributes of the data to reduce the variance within the group and increase the variance between the groups, which has been widely used in data classification when applying the GeoDetector method (lines 183-186).

  1. L171 "map" should be "maps".

Response: Changed as requested (line 191).

  1. Table 2 what is the difference between artificial grassland and natural grassland? Did you classify them in your classification maps? If yes, how about classification accuracy of them?

Response: Thank you for this comment. The grassland in this study is mainly natural grassland, and Table 2 is the reference standard for the grade value of LULC types. Artificial grassland is not involved in this study, and Table 2 has now been revised (line 244).

  1. Section 2.3.3 How did you implement the geographical detector method, with R package or what else?

Response: We used the Excel GeoDetector software developed by Wang et al. [4] to implement the geographic detector, which can be downloaded for free from the website (http://www.geodetector.cn) (lines 247-249).

  1. Figure 4. I would rather see the full error matrices rather than Figure 4.

Response: Following this comment, we have added the confusion matrix (lines 281-284).

  1. Figure 5. Compared with Figure 5a and 5b, there is a large expansion of construction land appeared in Figure 5c. Have you misclassified these patches since it is unreasonable to see such a large patch of urban expansion here?

Response: Thank you for the constructive comment. Based on these comments, we recalculate the data in Figure 5, and check the misclassified land types and modify them (line 296).

  1. It is difficult to understand Figure 6 without reading the text, please synthesis and simplify it to some specific topics instead of displaying all the numbers.

Response: We agree with the reviewer and synthesis it into some specific topics (lines 317-318).

  1. Figure 6 shows a lot of salt and pepper effect - which of these changes are real and what are the false changes?

Response: Thank you for pointing this out, and we have modified Figure 7 (lines 317-318).

  1. L281-286 Is nearly-ten year a reasonable time span to monitor the changes in land cover and land uses? Could you comment on that, especially when you are talking about the effect of grassland degradation may be caused by drought?

Response: We greatly appreciate the reviewer’s constructive comments. It is reasonable for us to monitor LULC changes in Gannan Prefecture for nearly ten years. This is because the study area from 2000 to 2009 was still dominated by traditional agriculture and animal husbandry. In order to pursue economic development, overgrazing of grassland led to grassland degradation. However, after 2009, the government noticed the seriousness of grassland and forest land degradation. The Chinese government implemented the Grassland Law and implementation of the Project of Grain for Green (http://lycy.gnzrmzf.gov.cn/info/1098/1023.htm). In addition, with the development of tourism in Gannan Prefecture, people's living standards improved since 2009, and LULC changes were obvious. Therefore, this study chose a time interval of 9 years to study the LULC change in this area.

  1. In discussion section, can you discuss about which land cover/land use classes are better classified? Also linked with the number of samples available from each class to train the classification.

Response: Thank you for this comment. Based on the suggestion, we conducted a detailed analysis in the discussion section (lines 366-379).

  1. L208 and L335 Estimating intensity of land cover/land use changes based on two dates is not reliable, how did you comment on this, please add it in the discussion section?

Response: Thank you for pointing this out. The land use intensity is the most intuitive performance of the human activity and can directly reflect the state of LULC. In this study, we use the land use degree index to directly describe the degree and intensity of land use in a specific period. We observe that the intensity of land use in Gannan Prefecture northeast is higher than that in the southwest. From 2000 to 2018, we can also see that the intensity of land use in Gannan Prefecture is gradually increasing, but the change is not significant, and it shows an aggregate distribution. Therefore, according to our research results, the intensity of land use in these two periods is reasonable and can reflect the current status of land use in Gannan Prefecture (lines 401-404).

  1. The discussion part is a little bit superficial without deep discussion about the driving factors of land cover/land use change. For instance, did these changing cases occur here similar as (or different from) other areas of the northeast of Tibetan Plateau? And how are these changes in other studies emphasized the similar drivers or different drivers, such as the economic shift and the rural-to-urban migration? And how were your results supported by other references? Cite references. The articles below focused on land cover/land use changes and drivers, may be good sources of evidence to enhance your discussion part:

Response: Thank you for the constructive suggestion. According to the reference paper provided by the reviewer, we discussed it in more detail in the resubmitted manuscript (lines 411-428).

References cited in the responses:

[1] Ge, J.; Meng, B.P.; Liang, T.G.; Feng, Q.S.; Gao, J.L.; Yang, S.X.; Huang, X.D.; Xie, H.J. Modeling alpine grassland cover based on MODIS data and support vector machine regression in the headwater region of the Huanghe River, China. Remote Sens. Environ. 2018, 218, 162-173, doi:10.1016/j.rse.2018.09.019.

[2] Ran, Q.W.; Hao, Y.B.; Xia, A.Q.; Liu, W.J.; Hu, R.H.; Cui, X.Y.; Xue, K.; Song, X.N.; Xu, C.; Ding, B.Y., et al. Quantitative Assessment of the Impact of Physical and Anthropogenic Factors on Vegetation Spatial-Temporal Variation in Northern Tibet. Remote Sens. 2019, 11, 22, doi:10.3390/rs11101183.

[3] Wang, M.Y.; Fu, J.E.; Wu, Z.T.; Pang, Z.G. Spatiotemporal Variation of NDVI in the Vegetation Growing Season in the Source Region of the Yellow River, China. ISPRS Int. Geo-Inf. 2020, 9, 17, doi:10.3390/ijgi9040282.

[4] Wang, J.F.; Li, X.H.; Christakos, G.; Liao, Y.L.; Zhang, T.; Gu, X.; Zheng, X.Y. Geographical Detectors-Based Health Risk Assessment and its Application in the Neural Tube Defects Study of the Heshun Region, China. Int. J. Geogr. Inf. Sci. 2010, 24, 107-127, doi:10.1080/13658810802443457.

Reviewer 2 Report

Li et al developed a land use product for Gannan Prefecture using Landsat remote sensing images and machine learning method on Google Earth Engine platform. This study also examined the trends of land use change and identified the driving factors of land use change. This paper has many interesting findings and fits the scope of Remote Sensing. I can understand the points that the authors want to make, but there are many grammar mistakes that need to be addressed. I would suggest this paper to be published after the authors address the comments below:

  1. Table 4: Could you explain how p value is calculated?
  2. It’s mentioned in the paper that q value indicates explanatory power of each power to land use change. However, it’s not clear to me whether it’s explaining the land use changes over time or spatial variations.
  3. Figure 6: It’s really hard to extract information from this figure with so many different colors. Please consider adding a figure showing the degree of land use.
  4. The language of this paper needs further edits to increase clarity. Below are a couple of examples:

Line 71: Delete “are”.

Line 74: change “Clarify” to “estimate”

Line 80: change “has” to ”have”

Line 166: remove “respectively”

Author Response

Dear Editors and Reviewers:

         Thank you for your letter and the comments concerning our manuscript entitled “Land use changes and driving factors in the northeast Tibetan Plateau: analysis based on a dense time stack of Landsat images and a geographical detector” (No: remotesensing-876172). These comments are all valuable and very helpful for improving our work and revising the manuscript. We have studied the comments carefully and revised the manuscript cogitatively. This letter is our point-by-point the details of the response to the comments raised by the reviewers’ comments. The revisions are marked in red in the manuscript. The responses to the reviewers are as follows:

Li et al developed a land use product for Gannan Prefecture using Landsat remote sensing images and machine learning method on Google Earth Engine platform. This study also examined the trends of land use change and identified the driving factors of land use change. This paper has many interesting findings and fits the scope of Remote Sensing. I can understand the points that the authors want to make, but there are many grammar mistakes that need to be addressed. I would suggest this paper to be published after the authors address the comments below:

  1. Table 4: Could you explain how p value is calculated?

Response: Thank you for this comment. In Table 4, we used the Excel GeoDetector software developed by Wang et al. [5] to implement the geographic detector, which can be downloaded for free from the website (http://www.geodetector.cn) (lines 247-249).

  1. It’s mentioned in the paper that q value indicates explanatory power of each power to land use change. However, it’s not clear to me whether it’s explaining the land use changes over time or spatial variations.

Response: We are grateful for the suggestions. Geographic detectors take into account the unique spatial differentiation of spatial data and quantify the contribution of various influencing factors to LULC change, and can effectively explain how each variable influences the spatial differentiation of land use. Therefore, we believe the q value in the article reflects the contribution of each driving factor to land use from space, and then we analyze the impact of each driving factor on land use in different years.

  1. Figure 6: It’s really hard to extract information from this figure with so many different colors. Please consider adding a figure showing the degree of land use.

Response: We really appreciate the reviewer’s instructive suggestions for our manuscript. We modify Figure 7 and synthesis it into some specific topics (line 318).

  1. The language of this paper needs further edits to increase clarity. Below are a couple of examples:

Response: We are very sorry for the language problems and the inconvenience caused. We have further improved the manuscript in terms of language and readability, and hope that it now meets the journal’s standard.

  1. Line 71: Delete “are”.

Response: We have modified it according to the comment (line 75).

  1. Line 74: change “Clarify” to “estimate”

Response: Changed according to the suggestion (line 78).

  1. Line 80: change “has” to ”have”

Response: Changed according to the suggestion (line 85).

  1. Line 166: remove “respectively”

Response: Changed according to the suggestion (line 181).

Reviewer 3 Report

The study area treated in this article is interesting, with several implications on the sustainable management of Tibetan Plateau ecosystems and landscapes. I congratulate the authors that decided to focus on this. Overall the article is structured in a form that can be proposed to the scientific community, however not to the Remote Sensing journal. Remote Sensing is seeking novel studies that are providing something really new to the remote sensing scientific community, some very relevant advances that can be applied in different contexts and with a certain evaluation of the accuracy of the methods proposed (therefore an accurate analisys on possible bias in the results is highly recommended). Unfortunately in the present article, there is a lack of such a novel message and also evaluation of the accuracy and bias in the final results. In addition, a method should be applied also to other study areas, in order to make it more robust and less affected by the weakness of the site-specific approach.

Based on the above critical issues, I would suggest the authors resubmit the article to another remote sensing journal that is accepting methods applied to specific case studies. I wish them great success in their interesting study.

Author Response

Dear Editors and Reviewers:

      Thank you for your letter and the comments concerning our manuscript entitled “Land use changes and driving factors in the northeast Tibetan Plateau: analysis based on a dense time stack of Landsat images and a geographical detector” (No: remotesensing-876172). These comments are all valuable and very helpful for improving our work and revising the manuscript. We have studied the comments carefully and revised the manuscript cogitatively. This letter is our point-by-point the details of the response to the comments raised by the reviewers’ comments. The revisions are marked in red in the manuscript. The responses to the reviewers are as follows:

Reviewer #3

       The study area treated in this article is interesting, with several implications on the sustainable management of Tibetan Plateau ecosystems and landscapes. I congratulate the authors that decided to focus on this. Overall the article is structured in a form that can be proposed to the scientific community, however not to the Remote Sensing journal. Remote Sensing is seeking novel studies that are providing something really new to the remote sensing scientific community, some very relevant advances that can be applied in different contexts and with a certain evaluation of the accuracy of the methods proposed (therefore an accurate analisys on possible bias in the results is highly recommended). Unfortunately in the present article, there is a lack of such a novel message and also evaluation of the accuracy and bias in the final results. In addition, a method should be applied also to other study areas, in order to make it more robust and less affected by the weakness of the site-specific approach.

Based on the above critical issues, I would suggest the authors resubmit the article to another remote sensing journal that is accepting methods applied to specific case studies. I wish them great success in their interesting study.

Response: Thank you for the constructive comment. The Tibetan Plateau is one of the world’s highest elevations, the harshest and most sensitive environment, and is well-known as the “Roof of the World” and the “Water Tower” of Asia. However, due to the complex terrain, there are great uncertainties among existing LULC products. Furthermore, most previous studies were concerned about the long-term change of vegetation in the TP. To date, there is still a lack of study on LULC classification in the Tibetan Plateau, and the LULC is usually based on low-resolution satellite imagery, such as AVHRR and MODIS. The high-resolution optical satellite images of the Tibetan Plateau are affected by the high cloud cover and data gaps. It is a challenging task to use a single scene image to monitor LULC changes on Tibetan Plateau.

        Given this background, we will answer the following questions: (1) This study application of dense time stacking of Landsat image on the GEE platform effectively overcomes the cloud cover challenge, which has proved to be a successful method to solve image quality problems. (2) The RF model can analyze the importance of characteristic variables, which improves classification accuracy while reducing data redundancy and processing workload. We use the explain( ) function on the classifier to view the importance of characteristic parameters for land use/cover (LULC) on the GEE platform. (3) The influencing mechanisms of LULC changes have been attracted much attention. However, traditional statistical methods such as correlation and regression are effective only when the relationships between LULC change and its driving forces are linear. These methods are subjective and ignore the spatial relationship between driving factors and LULC changes accurately, so it is difficult to study the underlying mechanism of their inherent changes. In this study, geographic detectors were used to quantify the impact of driving factors on the changes in LULC. This method considers the unique spatial differentiation of spatial data and quantifies the contribution of various influencing factors to LULC change, and can effectively explain how each variable influences the spatial differentiation of LULC. (4) Different from other developed regions, the driving factor analysis found that elevation was the most important factor affecting the intensity of land use in Gannan Prefecture. However, with economic development, the driving effects of anthropogenic factors on the intensity of land use have gradually expanded. In addition, livestock grazing activity is the most common factor influencing LULC changes in Gannan Prefecture. The study provides an important scientific basis for revealing the land use mechanism in the northeastern TP.
